# Peer J

# *Nephila clavata* L Koch, the Joro Spider of East Asia, newly recorded from North America (Araneae: Nephilidae)

E. Richard Hoebeke[1], Wesley Huffmaster[2] and Byron J. Freeman[3]

[1] Georgia Museum of Natural History and Department of Entomology, University of Georgia, Athens, GA, USA
[2] 124 Circle D Drive, Colbert, GA, USA
[3] Georgia Museum of Natural History and Eugene P. Odum School of Ecology, University of Georgia, Athens, GA, USA

## ABSTRACT

*Nephila clavata* L Koch, known as the Joro spider and native to East Asia (Japan, China, Korea, and Taiwan), is newly reported from North America. Specimens from several locations in northeast Georgia were collected from around residential properties in Barrow, Jackson, and Madison counties in late October and early November 2014. These are the first confirmed records of the species in the New World. Our collections, along with confirmed images provided by private citizens, suggest that the Joro spider is established in northeast Georgia. Genomic sequence data for the COI gene obtained from two specimens conforms to published sequences for *N. clavata*, providing additional confirmation of species identity. Known collection records are listed and mapped using geocoding. Our observations are summarized along with published background information on biology in Asia and we hypothesize on the invasion history and mode of introduction into North America. Recognition features are given and photographic images of the male and female are provided to aid in their differentiation from the one native species of the genus (*Nephila clavipes*) in North America.

Corresponding author
E. Richard Hoebeke,
rhoebeke@uga.edu

## INTRODUCTION

Golden orb-web spiders of the genus *Nephila* are pantropical in distribution (*Su et al., 2011*), exhibit extreme sexual size dimorphism (*Kuntner & Coddington, 2009*), and construct exceptionally large and impressive orb webs (*Kuntner, Gregoric & Li, 2010*). Until recently, the genus *Nephila* has been variously positioned by workers in either the family Araneidae or Tetragnathidae (see *Pan et al., 2004*). Some morphological, behavioral, and molecular evidence supports the elevation of the subfamily Nephilinae to family rank (*Kuntner, 2006*; *Harvey, Austin & Adams, 2007*; *Kuntner, Coddington & Hormiga, 2008*; *Su et al., 2011*; *Kuntner et al., 2013*), while other phylogenetic analyses support the inclusion of *Nephila* in the monophyletic family Tetragnathidae (*Dimitrov & Hormiga, 2009*). A recent study of the species composition of the Tetragnathidae and phylogenetic relationships of its

various lineages, including *Nephila* and its relatives (*Alvarez-Padilla et al., 2009*) concluded that nephilids are not closely related to tetragnathids as previously thought, that their position remains ambiguous, and that genetic analyses found Nephilidae to be sister to the Araneidae.

Since the early 1980s, accelerated global mobility of humans and their commodities, coupled with human-induced habitat destruction, alteration and fragmentation, has allowed countless species of exotic animals and plants to gain a foothold in distant locations (*Wheeler & Hoebeke, 2009*; *Mack et al., 2000*). Increasing trade with Asia and other Pacific Rim countries has favored the arrival and establishment of many new insects and other arthropods from the eastern Palearctic Region into North America. Spiders, largely due to their secretive habits and ability to hitchhike on various commodities, linked with the speed with which these items are transported to North American ports, can easily be introduced inadvertently into novel habitats. Detection of these immigrant species can sometimes be challenging and depends on the availability of taxonomists who can accurately identify them. In addition to the utilization of traditional morphological characters for species identification and differentiation, DNA barcoding (i.e., use of a short genetic marker—mitochondrial gene cytochrome oxidase I—in an organism's DNA) has become another important and universal tool in confirming the identity of animals not otherwise easily identifiable (*Hebert et al., 2003*; *Blaxter, 2004*; *Waugh, 2007*).

In this paper, we provide the first collection records and photographic evidence confirming the occurrence of the East Asian *N. clavata* L Koch in North America. Descriptive notes are given to facilitate the recognition of *N. clavata* in the Nearctic Region. We review literature on biology and habitat in its native range, and aligned COI sequences recovered in this study are compared with published sequences for *Nephila clavata.*

## MATERIALS AND METHODS

In late September 2014, one of us (WH) discovered a large, extremely colorful, but unrecognized female spider in an expansive web spun between upper tree branches about 2.5 m above-ground at a site in Madison County (Colbert), Georgia. The spider's identity was confirmed (by ERH) using taxonomic references on Asian spiders (cited herein), examining and comparing the male genital structures (palpal organ) with that illustrated in the primary literature, and by comparing specimens in hand with images posted on the Internet for this common eastern Palearctic species.

This remarkable discovery prompted us to find a way to determine if other people in northeast Georgia had encountered this large, attractive orb-weaving spider. Almost as soon as a press release appeared in a local newspaper in late October—calling attention to the first-time detection of this spider in North America with a color image—we heard almost immediately from several concerned citizens throughout a three-county area of northeast Georgia (*Lee, 2014*).

Over the course of the next week to 10-days, we had confirmed sightings from at least 9 locations (see locality data under North American records). With the exception of only a few sightings, we were able to collect specimens (females) from all locations. On one

occasion at a small community park (Braselton Park) in Braselton, Georgia (November 4), we (ERH and BJF) collected specimens (2 females, 2 males) in webs around the perimeter of a tennis court and along the wooded edge of the park. These are the only 2 males we have found to date.

A genetic analysis (by BJF) involved the sequencing of 2 female specimens from Jackson (GMNHTC 12242) and Madison (GMNHTC 12241) counties. Genomic DNA was extracted from single spider legs previously preserved in 95% ethanol, using a Chelex® 100 Resin and proteinase-k solution, incubated overnite at 55 °C (*Casquet, Thebaud & Gillespie, 2012*). The cytochrome C oxidase subunit I (COI) mitochondrial gene was amplified by polymerase chain reaction (PCR) using the primer combinations of LCO1490: 5′-GGTCAACAAATCATAAAGATATATTGG-3' and HCO2198: 5′-TAAACTTCAGGGTGACCAAAAAAATCA-3′ (*Folmer et al., 1994*; *Su et al., 2011*), and recovered a 639 base pair fragment with a starting motif of TTGGTACTGCAATAAGAGTA. PCR reactions followed conditions reported in *Wares, Gaines & Cunningham (2001)*. PCR products were assayed by electrophoresis using a 1.0% agarose gel and sequencing was performed by Macrogen (Macrogen, USA; available at https://www.macrogenusa.com/). Sequence files were aligned using CodonCode Aligner, and Geneious version 8.05 (created by Biomatters, available at http://www.geneious.com/) was used to visualize sequence data and compared with published sequences for Nephilidae.

Voucher specimens of *N. clavata* are deposited in the Collection of Arthropods, Georgia Museum of Natural History (UGCA), and specimens sequenced are also catalogued in the Georgia Museum of Natural History Genomic Collection (GMNHTC) University of Georgia, Athens, Georgia. In addition, an ArcGIS map of collection locations of *N. clavata* in northeast Georgia is provided.

## RESULTS

### *Nephila clavata* L Koch

Spiders of the genus *Nephila* are found throughout the tropical and subtropical regions of the world (*Kuntner et al., 2013*). According to the *World Spider Catalog (2014)*, the genus is comprised of 38 species and subspecies. However, *Kuntner (2005)* and *Kuntner & Coddington (2009)* note that of the 150 available species-level names, only 15 *Nephila* species are actually valid and known globally and that all recent descriptions probably represent synonyms.

In the New World, the genus is represented by only two extant species. *Nephila clavipes* (L.), often called the "banana spider" or the "golden silk spider," is found throughout Florida, the West Indies, as far north as North Carolina, across the Gulf States, through Central America, and into South America as far south as Argentina (*Weems & Edwards, 2004*). *Nephila sexpunctata* Giebel, endemic to the Neotropics, occurs in a limited range in southern South America (Brazil, Paraguay, Argentina).

*Nephila clavata*, native to East Asia, is referred to as the Joro spider in Japan (Japanese name: jorō-gumo) and the Mudang spider in Korea (Korean name: mudang gumi).

Literally translated, "jorō-gumo" means "entangling or binding bride" or alternatively "whore spider," while the Korean name translates to "shaman" or "fortune teller" spider.

**Descriptive notes**. *Nephila clavata* females can be readily distinguished from those of our native *N. clavipes* by their unique coloration (opisthosoma bright yellow when alive, with broad, horizontal bluish-green bands on the dorsum, large red markings on the venter, and long, black legs with yellow-orange bands). In contrast, the female of *Nephila clavipes* can be readily recognized by its color pattern alone—the silvery carapace, yellow spots on a dull orange to tan cylindrical opisthosoma, brown and orange-banded legs, plus the exaggerated hair brushes on the tibial segment of legs I, II, and IV (*Weems & Edwards, 2004*). Among other large orb-weaving spiders rivaling *N. clavata* in size, only the yellow and black garden spider, *Argiope aurantia* (F.), could be easily confused. However, it differs in body coloration (opisthosoma with striking yellow or orange markings on a black background) and in web architecture (circular web with a dense zigzag of silk, known as a stabilimentum, in the center).

The female (Figs. 1A–1C) of *N. clavata* can be characterized by the following combination of characters: Length: 17–30 mm; carapace of the prostoma dark brown with dense, golden to silvery-white vestiture; sternum dark brown with a trapezoidal yellow spot in the median area of the anterior half; legs black except for yellow-orange annulations at middle of tibia and at apical two-thirds of femur; base of metatarsus black in some specimens while yellowish at extreme base in others; tarsus varies from black to slightly orangish; palpi yellow-orange, with terminal segment black apically; opisthosoma ovoid-cylindrical with its dorsum (Fig. 1A) bright yellow (when alive) with five narrow to broad bluish-green horizontal bands; venter of opisthosoma (Fig. 1C) with anterior half marked with irregular black-brown oblique stripes and posterior half with two broad, oblique, red markings anterior to the spinnerets.

The male (Fig. 1D) is significantly smaller than the female, measuring only 4–8 mm in length. The prostoma is light brown with two dark brown longitudinal bands on both sides. The opisthosoma is elongate-oval with its dorsum greenish-brown with two yellowish longitudinal stripes on both sides of the dark brown median line. The palpal organ of the male is described and illustrated by *Feng* (*1990*: 96, Figs. 2–3); *Zhu, Song & Zhang* (*2003*: 76, Fig. 32, I, J); *Tanikawa* (*2007*: 95, Figs. 764, 766), and *Zhu & Zhang* (*2011*: 194, Fig. 136, I, J).

The anatomical characters given under descriptive notes above are adapted from the translation of *Kim, Kim & Lee (1999)*, *Tanikawa (2007)*, and *Zhu & Zhang (2011)*. Color images of the female, dorsal and lateral aspects, and the male are provided in *Yaginuma* (*1960*: 67, Plate 29, Fig. 163).

**Distribution.** *Nephila clavata* is found throughout Japan (except the island of Hokkaido), Korea, China, and Taiwan (*Kim, Kim & Lee, 1999*). The *World Spider Catalog (2014)* lists the species as occurring from "India to Japan".

**Material examined—New North American records** (Fig. 2). UNITED STATES: GEORGIA: *Barrow Co.*, Hoschton, 31-x-2014, C. Robbins, 5 females [34.089846, −83.8266]. *Jackson Co.*, Braselton, 31-x-2014, R. Barbani, 2 females & egg sac

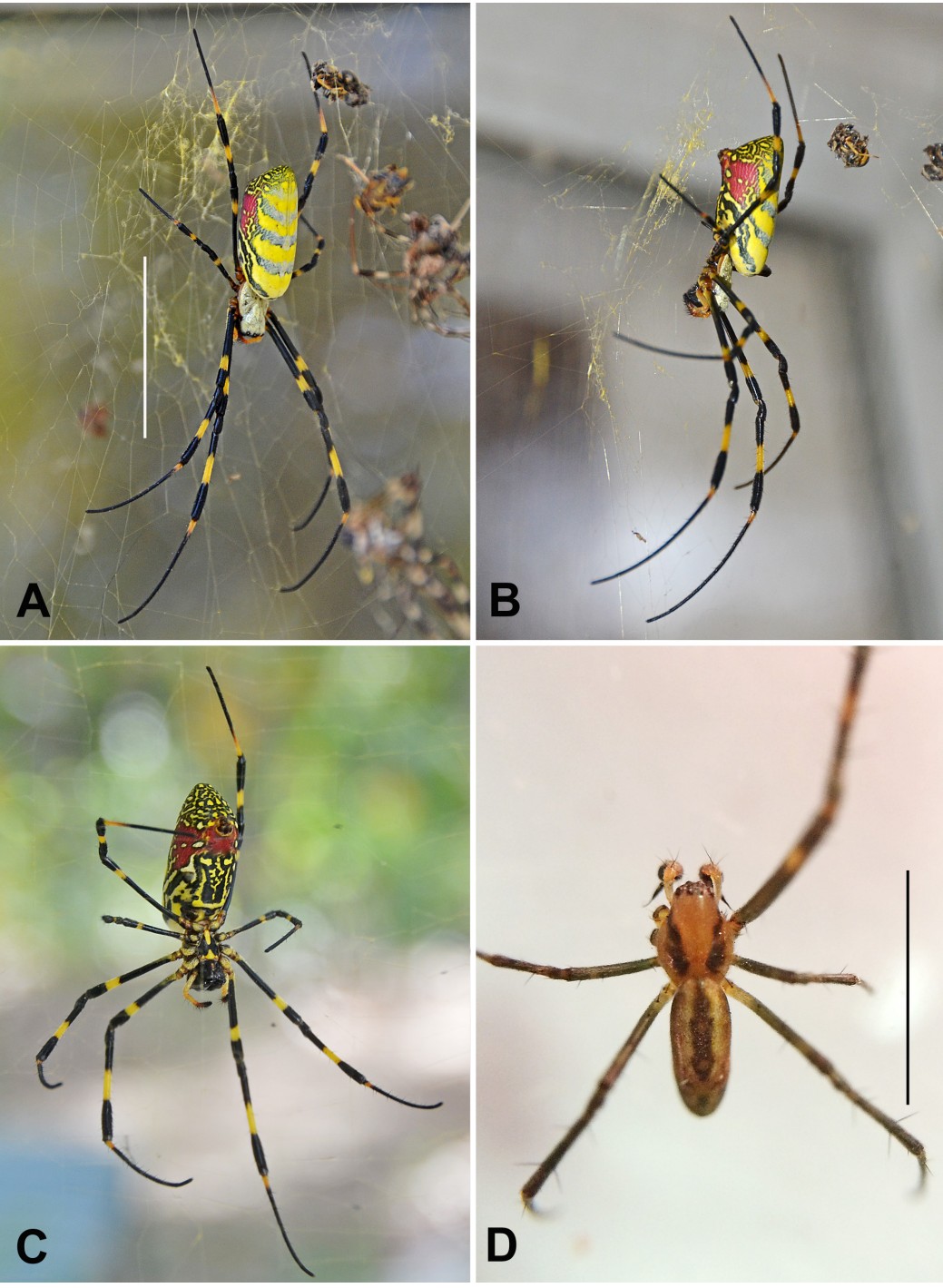

**Figure 1 Photographic images of *Nephila clavata* suspended in its web in northeast Georgia, taken in October 2014.** (A) female, dorsal aspect; scale bar, 30 mm. (B) female, lateral aspect. (C) female, ventral aspect. (D) male, dorsal aspect; scale bar, 5 mm. Photos A–C were taken in Hoschton, Georgia by Jeremy Howell; photo D was taken in Braselton, Georgia by BJ Freeman.

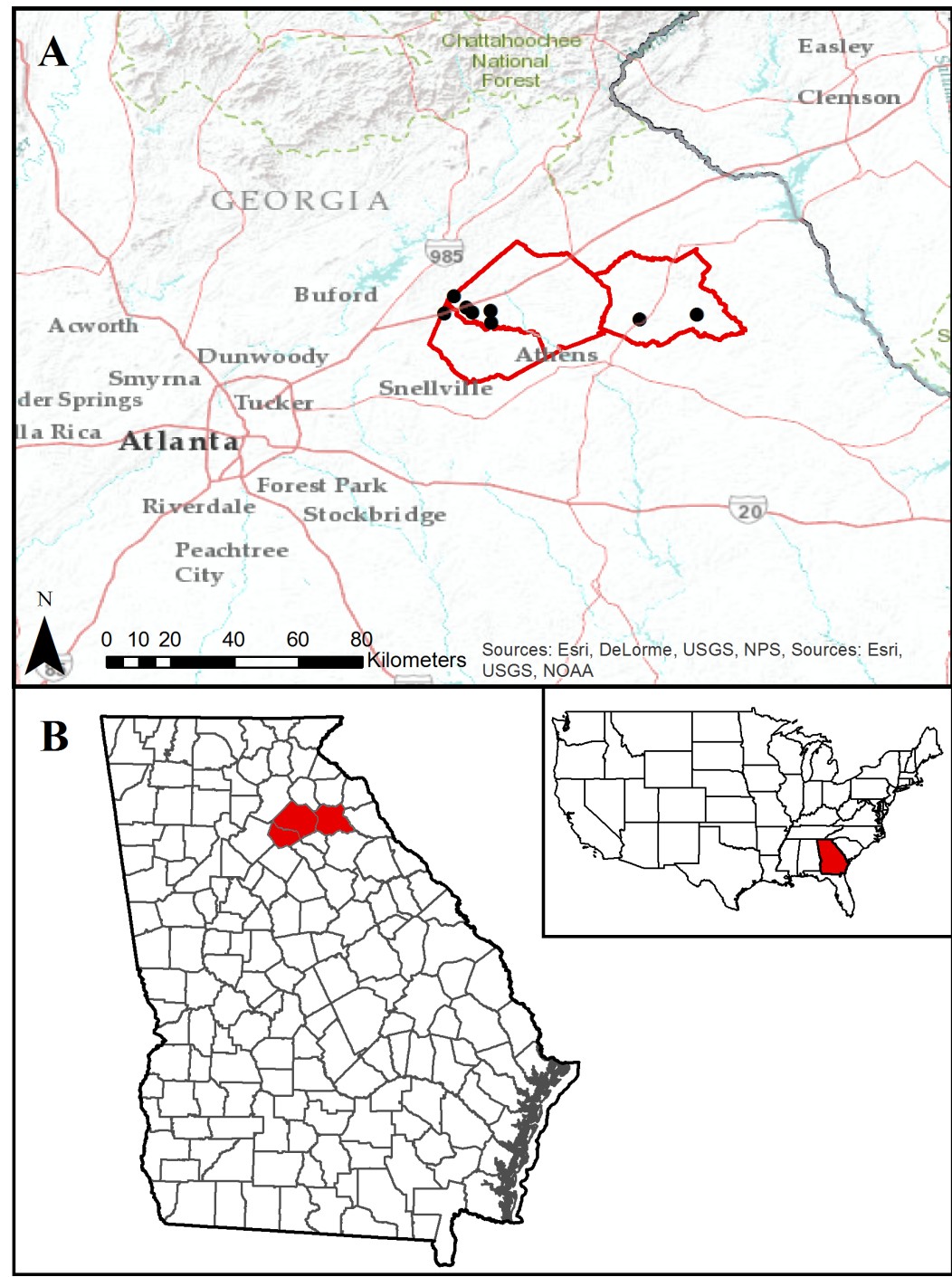

**Figure 2 Map of Georgia, USA.** (A) geocoded localities in Barrow, Jackson, and Madison counties, about 64 km northeast of Atlanta, where collections and sightings of *Nephila clavata* were made. (B) location of the above-mentioned counties in northeast Georgia.

[34.138962, −83.798877]; Braselton, Braselton Park, 4-xi-2014, GMNHTC 12242, E. R. Hoebeke and B. J. Freeman, 2 females & 2 males [34.1072909, −83.7638141]; Braselton, 9-xi-2014, K. Youngblood (image only) [34.142209, −83.7564699]; Hoschton , 4-xi-2014, C. Glick (image only) [34.064256, −83.693733]; Hoschton, 4-xi-2014, K. and J. Howell, 1 female [34.093613, −83.747015]; Jefferson, 4-xi-2014, C. Hamilton, 1 female [34.097902, −83.6953489]. *Madison Co.*, Colbert, 4-x-2014, GMNHTC 12241, W. Huffmaster, 1 female [34.073869, −83.2742425]; Comer, 7-xi-2014, K. Fields, 1 female [34.0878353, −83.1133751].

**Bionomics and habitat**. *Nephila clavata* is univoltine in Japan. Eggs overwinter and spiderlings emerge from the egg cocoons in early June in central Japan (*Miyashita, 1986*; *Miyashita, 1990*; *Miyashita & Hayashi, 1996*). Males reach maturity by late August, and females become sexually mature in September and early October. After mating, oviposition occurs from mid-October to November, resulting in the production of only a single egg sac (*Miyashita & Hayashi, 1996*). Adult males do not construct their own webs but join females on their webs (*Miyashita, 1994*; *Miyashita & Hayashi, 1996*). Females produce large, golden-yellow, basket-shaped webs between tree branches. Females oviposit between 400–500 eggs in a single egg sac (*Kim, Kim & Lee, 1999*), inside a densely woven silk cocoon that is attached to the bark of trees, on leaves, or upon other human-made structures. *Harvey, Austin & Adams (2007)* described the production of a non-viscid, dense, silk platform upon which an egg sac is deposited, followed by the placement of a thick layer of bright yellow, loose, flocculent or looped silk over the egg mass. In Korea, spiders are still active until late November. All adults die off with the onset of winter. *Kim, Kim & Lee (1999)* noted that this species inhabits mountainsides, fields, or is found in urban and non-urban woodland sites in its native habitat. In Japan, spiders are commonly found throughout lowland forests and can also inhabit building spaces in the vicinity of small urban woods (*Miyashita, 1990*), especially where there is a high degree of human disturbance.

In northeast Georgia, large, mature females were first observed beginning in late September and persisted until mid-November when temperatures began to cool significantly. Most spiders were found in large webs attached to the exterior of homes near porch lights, on wooden decks, or among shrubs and flowering bushes near to homes. At one location in early November, spiders were found in webs attached to small trees and high weeds along a wooded area of a small city park (Braselton Park, Braselton). Here we also found two males in a web with a single female. At one Braselton home site, we found an egg sac, covered by a dense cocoon of silk and guarded by a female, attached to vinyl siding of a home.

**DNA barcoding**. The two COI sequences recovered in this study were aligned with GenBank Popset (537470784) consisting of forty-four taxa of Orbiculariae (*Kuntner et al., 2013*) and eight additional sequences for *N. clavata* retrieved from Genbank (*Su et al., 2011*; *Arabi et al., 2012*), and unpublished. A phylogeny (Fig. 3) based on COI sequence data was performed using Mr. Bayes 3.2.2 (*Huelsenbeck & Ronquist, 2001*) as implemented in Geneious 8.05. The outgroup was the European cave spider *Meta menardi* (Latreille).

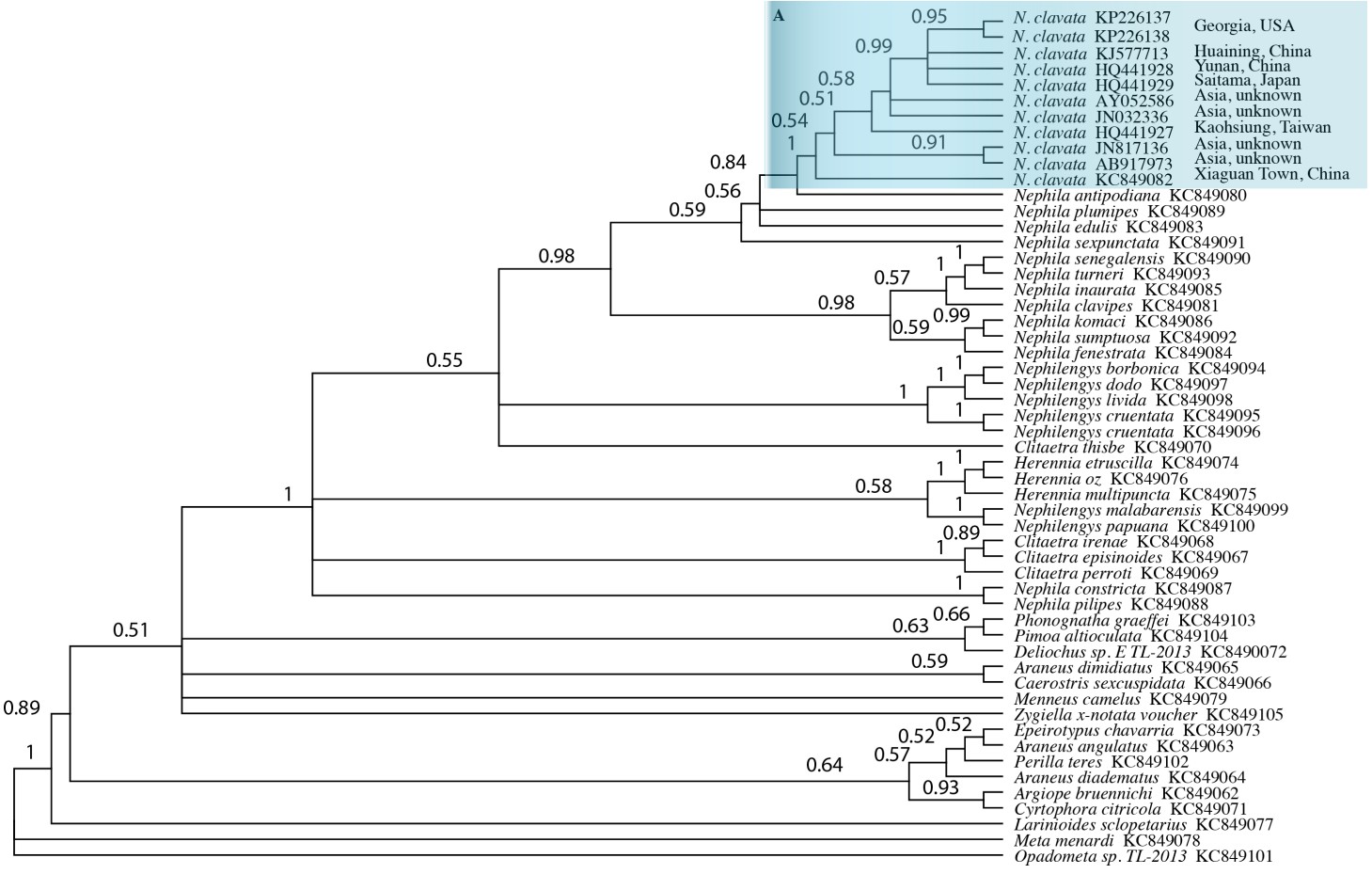

**Figure 3 Bayesian phylogeny for 52 individuals of Orbiculariae.** Bayesian phylogeny based on COI sequences for 52 individuals of Orbiculariae. The outgroup is *Meta menardi*. Posterior probabilities are presented at each node. Species names are followed by GenBank numbers. The *Nephila clavata* clade is shaded blue and labeled A with the country of origin following each sequence.

Mr. Bayes parameters included the HKY85 substitution model, a run length of 1,100,000 generations, subsampling frequency of 200 and burn-length of 100,000. All *Nephila clavata* sequences were recovered in a monoplyletic clade with a posterior probability of 1.0. A comparison of aligned sequences for *N. clavata* revealed two groups with differing haplotypes at positions 309 and 318 with this alignment motif. One group, represented by GenBank accession numbers: KC849082, AY052586, JN032336, HQ441927, JN817136, and AB917973, has Adenine at position 309 and Guanine at position 318, and the second group represented by KJ577713, KP226137, KP226138, HQ441928 and HQ441929 has Guanine at position 309 and Thymine at position 318. Sequences from Georgia specimens GMNHTC 12241 (GenBank KP226137) and GMNHTC 12242 (GenBank KP226138) were identical.

## DISCUSSION

Hundreds of non-native species of animals and plants have been inadvertently introduced into managed landscapes and natural ecosystems of North America. In fact, an estimated

2,000+ species of insects and arachnids have become established in North America over the past half-century and are largely attributable to a dramatic increase in travel and international commerce (*OTA, 1993*). To date, approximately 60 species of non-native spiders (Araneae) have been detected in North America, with the majority originating from Europe and Asia (a species list of adventive spiders in North America is posted at http://bugguide.net/node/view/32329#Anchor_Araneae). *Nephila clavata* becomes the newest species to be added to this list of non-native spiders occurring in North America.

**Potential pathways of introduction.** Accidental human transport of spiders and their egg masses on or within cargo containers, on plant nursery stock and on crates and pallets (*Nentwig & Kobelt, 2010*), can explain and account for the presence of many European and Asian species in North America. We think this is the probable means of transport by which *N. clavata* gained entry. If accidental transport of *N. clavata* were to occur in late August to early October from countries of origin in East Asia, then the spiders' reproduction would be at its height and there would be a greater likelihood that egg sacs might be deposited on structures or plant material being exported.

**Country of origin**. Sequence data for the COI gene recovered from these Georgia specimens confirms our identity of *N. clavata* by comparison with published sequences from nine different Asian populations of *N. clavata*, and suggests that specimens recorded from Georgia are more closely related to sources in China or Japan, rather than Taiwan, based on the unique haplotypes. The Georgia specimens sequenced span the geographical range of our records, suggesting that they share the same source. The Yunan (HQ441928) and Huaining (KJ577713) China specimens are over 1,500 km distant from each other, and the Saitama, Japan population (HQ441929) is over 4,000 km from Yunan, and all of these sequences are identical for about 700 base pairs of COI and share the same variable haplotypes as the Georgia specimens. Specimens from Xiaguan, China (KC849082), Kaohsiung, Taiwan (HQ441927), and unspecified Asian locations (AY052586, JN032336, JN817136, and AB917973) share a different set of variable haplotypes from the Georgia specimens, and also span a similarly broad geographic range.

**Dispersal opportunities.** Once spider populations have successfully colonized in a given location, individual spider movement might be best explained by the movement of transport vehicles along major rail or road corridors (*Nentwig & Kobelt, 2010*) or possibly by ballooning of spiderlings in the spring after egg hatch. Ballooning is a behavior by which spiders use air-borne dispersal to move between locations. Depending on mass and posture, a spider might be taken up into upper air streams (*Suter, 1992*), while its aerial movement would be dependent on convection air currents and on the drag of the silk parachute (*Greenstone, Morgan & Hultsh, 1987*). Members of the genus *Nephila* have been suggested by some (*Kuntner & Agnarsson, 2011*; *Su et al., 2011*) to be among the best spider dispersers via their ballooning behavior.

A preponderance of sightings and collection of specimens of *N. clavata* were centered on a restricted area in and around Braselton and Hoschton, Georgia. One property owner in Hoschton indicated that the spider had been present around her home for at the past 4 years. We are not necessarily suggesting that this area represents the probable arrival

point of this Asian spider, but it could be argued that the industrial and business history of the region might demonstrate it to be a possibility. The town of Braselton is a thriving business location on the I-85 business corridor, located 64 km northeast of Atlanta. As such, its location on the I-85 corridor provides excellent transportation access. It is home to many warehousing and distribution facilities that transport containerized freight from overseas.

Collection locales in Jackson and Barrow counties ($n = 7$) are clustered on the I-85 corridor but the Madison county records ($n = 2$) are located in a rural mixed farm landscape, not adjacent to commercial transportation corridors. The Madison county sites are ca. 50 km due west and downwind from the other Barrow and Jackson county sites. We hypothesize that these downwind sites were colonized by aerial dispersing spiderlings using the prevailing westerly winds and suggest that other populations might be found along this route.

## ACKNOWLEDGEMENTS

We are indebted to the following individuals who alerted us to the presence of *N. clavata* around their homes in northeast Georgia; their e-mails, with attached digital images of the spider, allowed us to determine the validity of their sightings: Ronald Barbani and Kelly Youngblood (Braselton); Christine Robbins, Kristin and Jeremy Howell, and Crystal Glick (Hoschton); Crystal Hamilton (Jefferson); and Kathy Fields (Comer). We also acknowledge Gang Hua and Qi Zhang (Department of Entomology, University of Georgia) and Tae-young Lee (Entomology, UGA) for helping with the translation—Chinese to English and Korean to English, respectively—of taxonomic passages in the Asian spider literature; Kent Loeffler (Chico, California) for assistance with digital image enhancement, editing, and layout for Fig. 1; and Mary Freeman (U.S. Geological Survey, Athens, Georgia) and Carrie Straight (USFWS, Athens, Georgia) for generating the ArcGIS map of known spider localities in Fig. 2. We also thank the Wares Lab (Department of Genetics, University of Georgia) for advice, engaging discussions, and use of equipment, as well as Matjas Kuntner (Slovenian Academy of Sciences and Arts, Ljubljana, Slovenia) and two anonymous reviewers for valuable suggestions that improved an early draft of the manuscript.

### Funding

Support for analyses was provided by the Georgia Museum of Natural History internal funds. The funders had no role in study design, data collection and analysis, decision to publish, or preparation of the manuscript.

### Grant Disclosures

The following grant information was disclosed by the authors:
Georgia Museum of Natural History.

## Competing Interests

The authors declare there are no competing interests.

## Author Contributions

- E. Richard Hoebeke and Byron J. Freeman conceived and designed the experiments, performed the experiments, analyzed the data, contributed reagents/materials/analysis tools, wrote the paper, reviewed drafts of the paper.
- Wesley Huffmaster conceived and designed the experiments, performed the experiments, reviewed drafts of the paper.

## DNA Deposition

The following information was supplied regarding the deposition of DNA sequences:
GenBank KP226137-KP226138.

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
