# Peer review of "Nephila clavata L Koch, the Joro Spider of East Asia, newly recorded from North America (Araneae: Nephilidae)"

_PeerJ, doi:10.7717/peerj.763_

## Round 0.1 · original submission · Major Revisions

Dear authors,

Having read the submitted reviews, your manuscript requires major revisions prior to being considered for acceptance. Please carefully read through all three reviews.

Please note the citations to websites such as Wikipedia are not suitable. When possible use primary, or if necessary secondary, literature.

Finally, discussion regarding the phylogenetic position and origins of the North American spiders should be carried out in a phylogenetic context. Reviewer 1 mentions GenBank sequences that could be of use. Also, please double-check your sequences are up on GenBank, as reviewer 1 mentions not being able to find them.

·

Basic reporting

The manuscript reports on colonies of an Asian spider, Nephila clavata, in USA. The finding is certainly unusual and as such probably worth reporting. However, the results of this note are presented in a taxonomic order of subheadings, but this contribution most certainly is not a taxonomic paper. It lacks many important features to be taxonomic, such as anatomical details, overview of other Nephila species features, etc. Hence, the main suggestion I have is to rewrite that part of the results. This paper should not be taxonomic, hence, take out the synonymy, and retitle the diagnosis and the description into something like Regional species diagnosis, and Comments or similar. I made comments on how to rewrite them in appropriate pdf sections.

Experimental design

Unfortunately, the only scientific part of this paper, the barcoding result, is inconclusive. The discussion of the paper is thus at best speculative, as the source population could be China or Japan. There are other sequences of N. clavata available on GenBank including my own from China. However, I cannot find the sequences that the authors have submitted to GenBank to examine them.

Validity of the findings

I have no doubt even form only the photographs, that this is indeed Nephila clavata. But again, please take out of this ms taxonomic components to make it publishable. Also, referring to websites in discussing species biology is not scientific practice.

Additional comments

I made numerous further smaller suggestions for improvement in the ms file. But overall, please restructure the ms to conform to a note reporting on an unusual finding rather than attempting to provide a taxonomic treatment.

Reviewer 2 ·

Basic reporting

The present ms has a short and concise introduction that is very focused on the family nephilidae. There is a note on the recent erection of this family but there are few papers (not cited here) that have stated that this may be preliminary and have shown that Araneid placement may be more adequate.
Because the main point in the ms is to report a newly introduced species I miss seeing some introduction on the topic of introduced species (spiders or other). This seems certainly relevant here.
I also feel that the structure of the ms may not be best for its purpose. Since you are not re-describing the species here I am surprised to see this format for the systematics findings. I would suggest that you change this part of the text (diagnosis and description). You may want to discuss what are the characters that make you think this is N. clavata and then perhaps describe the morphology (and variation) of your specimens. The same goes for the biology section. May be you want to discuss what you observed for your specimens and compare whit what is known instead of just summarizing published research.
Finally in the discussion you talk about potential country of origin, dispersal routes, etc but what about potential impact? I understand that it may be difficult to quantify what is the impact of this introduced species on local biota but may be you could hypothesize about it. Also, what about potential measures to contain the spread of the species in North America?

Experimental design

In the material and methods section you should also provide some information on the following:
How sequences similarity was measured?
Did you use any phylogenetic or NJ analyses to place the two sequences?
This is scattered throughout the text now.

Validity of the findings

Given the presented evidences and the methods used I do not have specific comments here.

Reviewer 3 ·

Basic reporting

No Comments

Experimental design

There is no experimental design. This note documents the discovery of an introduced species

Validity of the findings

The discovery of the species in N America is interesting, but the article is written in a ´chatty´ manner, should be a simple description of the findings.

Additional comments

This is a short note documenting the surprising discovery of an introduced Asian Nephila spider, having apparently established a population in N America.

The paper can be improved. There is no need to offer a ´chatty´description of the circumstances of how the spider was found in the field, photographed with a cell phone, identified with an internet search etc. Simply state its discovery. location, and scientific resources used to identify it.

I do not see why the discovery of an introduced species requires a (superficial) review of its systematics, such as synonyms. This has nothing to do with the findings of the paper and proper review of synonyms and systematic status of hte species should be left to the expert. I don´t either see why a ´formal´description is needed as these are available from recent literature. Also why is a recap of its biology in Japan necessary - for example, why is this a venue to discuss correlation between fecundity and body size?

If the authors wish to discuss possible routes of colonization they should do formal phylogenetic analyses using the DNA data at hand, rather than just ´blast´ followed by speculation

---

## Round 0.2 · Minor Revisions

Dear authors,

Your revised version of the manuscript has been re-reviewed, and the general consensus is that it is much improved over the original submission. Reviewer 2 has some minor comments that should be looked into, and reviewer 3 suggests adding the evolutionary tree in as a supplementary figure. I agree that the inclusion of your phylogram should be made, and that instead of it being a supplementary figure, have it added as your final figure.

Once these minor suggestions have been made, I forsee no obvious reason your manuscript it should not be acceptable for publication.

Yours sincerely,
Mark Young

Reviewer 2 ·

Basic reporting

The authors have followed most of the recommendations from the reviewers and the current version is greatly improved. I have just few minor comments at that point:
1) I still feel that the introduction needs some work. For example it just focuses on introduction to the family but topics of introduced species and barcoding identification are not touched upon. However, they are central for the discussion. Given the extensive introduction on the family I feel that is also necessary to add that elevation of the group to family rank is still contentions (for example this is copied from the world spider catalog literally: N.B.: elevated to family level and relimited by Kuntner, 2006: 24, both actions regarded as dubious by Dimitrov & Hormiga, 2009: 21 (see also Álvarez-Padilla et al., 2009).
2) I understand the reason for keeping a diagnosis here to aid identification in North America. However, the term has a specific connotation in systematics context so it may be better to change the heading or refer to the original diagnosis of the species and then explain that here it is amended with notes relevant to the identification of the species in North America.

Experimental design

no comments

Validity of the findings

no comments

Reviewer 3 ·

Basic reporting

The manuscript has been greatly improved. I would like to be able to examine the phylogenetic tree - perhaps provide that as supplementary Figure?

Experimental design

No Comments

Validity of the findings

Interesting

Additional comments

This paper has been revised and greatly improved. I think it is publishable as is, with phylogenetics figure made available as supplementary data

---

## Round 0.3 · accepted · Accept

Thank you for the swift revision of your manuscript, and I am delighted to accept it for publication at PeerJ.